# Radiative Regime According to the New RAD-MSU(BSRN) Complex in Moscow: The Roles of Aerosol, Surface Albedo, and Sunshine Duration

**Daria Piskunova** [1,*], **Natalia Chubarova** [1], **Aleksei Poliukhov** [1,2] **and Ekaterina Zhdanova** [1]

[1] Faculty of Geography, Moscow State University, Leninskie Gory, 119991 Moscow, Russia; chubarova@geogr.msu.ru (N.C.); aeromsu@geogr.msu.ru (A.P.)

[2] Moscow Center for Fundamental and Applied Mathematics, Leninskie Gory, 119991 Moscow, Russia

* Correspondence: piskunovada@my.msu.ru

**Abstract:** The radiative budget is one of the key factors that influences climate change. The aim of this study was to analyze the radiative regime in Moscow using the RAD-MSU(BSRN) complex and to estimate the radiative effects of the main geophysical factors during the 2021–2023 period. This complex is equipped and maintained according to the recommendations of the Baseline Surface Radiation Network; however, it is not a part of this network. In cloudless conditions, the decrease in global shortwave irradiance (Q) is about 18–22% due to the aerosol content with a pronounced change in the direct to diffuse ratio. In winter, the increase in Q is about 45 W/m$^2$ (or 9%) at h = 30° due to a high surface albedo and reduced aerosol and water vapor contents, while the net shortwave irradiance (Bsh) demonstrates a significant decrease due to the prevailing effects of snow albedo. In cloudy conditions, a nonlinear dependence of Q and Bsh cloud transmittance on the relative sunshine duration is observed. The mean changes in Q for the 2021–2023 against the 1955–2020 period are characterized by negative anomalies (−22%) in winter and positive anomalies in summer (+3%) due to the changes in cloudiness. This is in line with the global tendencies in the long-term changes in shortwave irradiance in moderate climates in Europe in recent years.

**Keywords:** shortwave irradiance; longwave irradiance; sunshine duration; surface albedo; aerosol; cloudiness; Baseline Surface Radiation Network (BSRN); CIMEL sun photometer; Kipp & Zonen

## 1. Introduction

The radiative budget is known to be the main regulator of the Earth's climate. Despite the great progress in recent studies [1], the uncertainties of its evaluation are still noticeable [1,2]. To attribute possible scenarios of climate change, one needs accurate assessment of shortwave and longwave radiations and their changes in different geographical regions of the world.

The measurements of the Baseline Surface Radiation Network (BSRN) [3] employed since 1992, are widely used for the analysis of the radiative regime in different geographical regions, the validation of satellite radiative retrievals, and the testing of radiative transfer models and re-analysis data [2,4–8]. Currently, the BSRN includes a relatively small number of stations (currently, 51 active sites), which are located in different climatic zones from 80° N to 90° S. Some stations are not BSRN members but use the BSRN approach in their studies [9–11].

According to the BSRN, the detailed analysis of radiative regimes is given for Namibia [12], the Netherlands [13], and Spitsbergen [14]. The long-term measurements of BSRN observations are also used for detecting global dimming and brightening effects [2,15,16].

Special attention has been paid to the application of the BSRN data for the verification of the results of CMIP project models [2,4]. However, there is still no complete consistency

in the model estimations of radiation fluxes [4] and observations. In this regard, the continuation and spread of high-quality BSRN observations are of great importance [17].

For better understanding of the physical mechanisms of the variability in the radiation budget, the role of the main geophysical factors, like aerosols, cloudiness, and surface reflectance, should be accurately quantified. The estimations of cloud effects on solar radiation are given in [13,18]. The influence of atmospheric aerosol is discussed in [19,20], while the effects of surface albedo are analyzed in [14,21]. However, due to large geographical variety, there is still considerable uncertainty when attributing the effects of different geophysical factors on radiation.

The Meteorological Observatory of Lomonosov Moscow State University (MSU MO, 55.707° N, 37.52° E, H = 192 m) has provided a long series of radiative measurements since 1954 [22]. The observations include measurements of direct, diffuse, global, and reflected shortwave irradiances, and net radiation using thermoelectric Russian instruments recommended by the Roshydromet agency [23], as well as UV irradiance (300–380 nm) and biologically active erythemally weighted UV irradiance [24]. The MSU MO is a part of the national and international radiative networks. The radiative data of the MSU MO are stored in the database of the World Radiation Data Center (WRDC) [25]. The analysis of long-term radiative measurements, as well as the main features of the radiative regime of Moscow, are described in [24,26,27]. In addition, the measurements of the MSU MO are used for testing reconstruction models [28].

In order to improve the quality of radiative observations and to meet the standard of high-quality measurements, a new radiation RAD-MSU(BSRN) complex was installed at the MSU MO in summer 2021 [29]. This complex is equipped with high-quality Kipp & Zonen instruments [30], which are recommended for the application at BSRN stations. However, currently, the MSU MO is not part of the BSRN.

The aim of this paper was to analyze the main features of radiative characteristics on the basis of the data of the new RAD-MSU(BSRN) complex for its first two-year period. In the analysis, we also focused on the effects of aerosol content and cloudiness, and the influence of the surface albedo on shortwave irradiance. In addition, we performed comparisons between the RAD-MSU(BSRN) data and ongoing standard radiative measurements, and estimated the radiative changes in recent years compared to the long-term observations at the MSU MO.

## 2. Materials and Methods

### 2.1. The Description of the RAD-MSU(BSRN) Complex

The RAD-MSU(BSRN) complex was installed at a height of about 10 m on the roof of the MSU MO (Figure 1) for providing measurements of downwelling irradiance. The upwelling irradiance was measured over a natural surface (grass and snow, depending on the season) by the instruments located at ground level nearby [29]. Downwelling radiative measurements included direct (S), diffuse (D), global shortwave irradiance (Q), and downwelling longwave irradiance (L_U), UV-A (315–400 nm) irradiance, erythemally weighted UV irradiance (ER), and sunshine duration (Sd) (Table 1). Upwelling radiative measurements included reflected shortwave irradiance (R) and upwelling longwave irradiance (L_L). The RAD-MSU(BSRN) was maintained following the recommendations and guidelines of the BSRN [31,32] and was equipped with a full set of instruments recommended by the BSRN [8,31,32].

The instruments of the new complex were installed within a few meters of the standard MSU MO instruments.

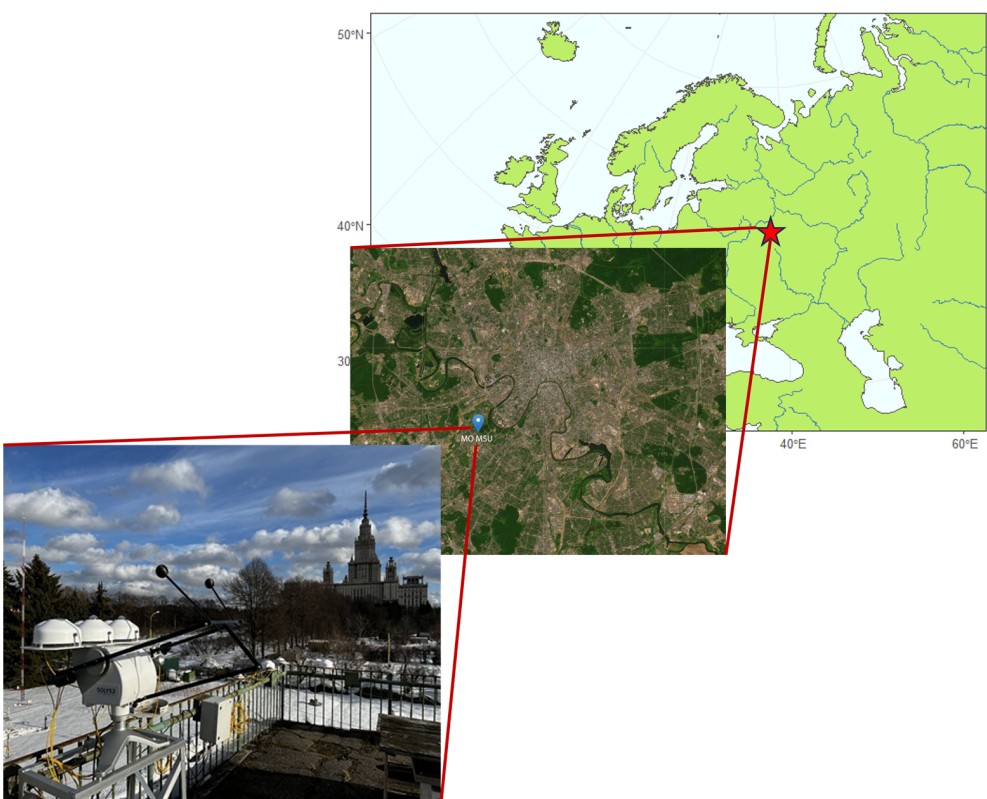

**Figure 1.** The photo of the RAD-MSU (BSRN) complex on the roof of the MSU MO (55.707° N, 37.52° E, H = 192 m), and its location on the satellite image (ESRI World Imagery) and on the map.

**Table 1.** Observations and set of instruments of the RAD-MSU(BSRN) complex [30].

| Parameters | Designations | Instruments | Measurement Errors |
|---|---|---|---|
| Direct normal shortwave irradiance | S | CHP1 Pyrheliometer | <0.5% |
| Diffuse shortwave irradiance | D | CMP21 Pyranometer | $<\pm 10 \text{ W/m}^2$ |
| Global shortwave irradiance | Q | CMP21 Pyranometer | $<\pm 10 \text{ W/m}^2$ |
| Downward longwave irradiance | L_U | CGR4 Pyrgeometer on the roof | <1% |
| Reflected shortwave irradiance | R | CMP21 Pyranometer | $<\pm 10 \text{ W/m}^2$ |
| Upward longwave irradiance | L_L | CGR4 Pyrgeometer on the ground | <1% |
| Ultraviolet irradiance in the range of 315–400 nm. | UVA | SUV-A UVA Radiometer | $<\pm 5\%$ |
| Erythemal UV irradiance | ER | SUV-E UVE Radiometer | $<\pm 5\%$ |
| Sunshine duration | Sd | CSD3 Sunshine Duration Sensor | >90% (monthly sunshine hours |

Specialized data processing software was developed [29]. This software included the correction of shortwave irradiance on zero offset, the estimation of several important parameters including surface albedo, shortwave, longwave, and total net irradiances, as well as the calculation of solar elevations. Special attention was paid to the evaluation of automatic quality control flags and their incorporation in the software. Several quality flags were applied according to recommendations [31]. As a quality flag, we also used the ratio of the measured global shortwave irradiance to the global shortwave irradiance, estimated as the sum of diffuse irradiance and direct irradiance on a horizontal surface. The detailed description of the different quality check procedures is presented in [29].

The comparisons with the standard radiative measurements are shown in Table A1. On average, there is a good agreement between the two datasets. The differences in annual doses for direct, diffuse, and reflected irradiances do not exceed 2.5%, and for global

shortwave irradiance, the difference is even smaller than 0.9%. However, one can see a larger deviation during the cold period. The most significant differences are observed for diffuse and reflected irradiances in January (higher than 10%). They can be explained, to some extent, by the instrumental uncertainty of standard Russian instruments. For reflected irradiance, the deviation can be also attributed to slightly different locations of the instruments on the territory of the MSU MO, which may provide some variations in surface albedo. However, even the largest differences for diffuse and reflected irradiances lie within the uncertainty of measurements by standard Russian instrumentation [23]. During summer conditions, at high solar elevations, the agreement between the two datasets is much better (see Table A1). However, in summer, there is also a noticeable negative difference of about 3–9% for reflected irradiance. In future, we plan to continue the study of the observed discrepancies using more statistics.

We analyzed radiative characteristics over the two-year period from 1 September 2021 to 31 August 2023 (Table A2). The hourly, daily, and monthly datasets were generated using one-minute resolution data. For analyzing the effects of the geophysical factors, the radiative data were normalized to the mean Sun–Earth distance.

For the estimation of the seasonal anomalies in 2021–2023, we also used the database of standard radiative measurements over the 1955–2020 period [32]. In addition, we used several standard meteorological parameters: visual cloud observations, percent of snow cover at the site, and snow height.

### 2.2. The Description of the Procedure for Estimating Aerosol Characteristics

Using the collocated aerosol measurements of the CIMEL CE-318 sun photometer, we estimated the aerosol optical thickness ($\tau_{aer\,\lambda,i}$) at different wavelengths and Angstrom exponent, following the methods developed in the third version of data of the Aerosol Robotic Network (AERONET) [33,34]. Unfortunately, during this period the application of the direct AERONET retrievals was impossible due to some formalities in maintaining the AERONET at the MSU MO. The estimation of aerosol optical thickness was made for 340, 380, 440, 500, 675, 870, and 1020 nm wavelengths using the following equation:

$$\tau_{(aer\,\lambda,i)} = \ln\left(\frac{S_\lambda}{S_{0\lambda}}R_i\right)\frac{1}{m} - \tau_{(H2O\,\lambda,i)} - \tau_{(O3\,\lambda,i)} - \tau_{(NO2\,\lambda,i)} - \tau_{(CO2\,\lambda,i)} - \tau_{(CH4\,\lambda,i)} - \tau_{(rel\,\lambda,I)}, \tag{1}$$

where $S_\lambda$ is the spectral direct irradiance at wavelength $\lambda$, $S_{0\lambda}$ is the extraterrestrial spectral irradiance in relative units given in Table A3, $R_i$ is the correction on the Sun–Earth distance at i-day, m is the optical mass of the atmosphere, $\tau_{(H2O\,\lambda,i)}$ is the optical thickness of water vapor, $\tau_{(O3\,\lambda,i)}$ is the optical thickness of ozone, $\tau_{(NO2\,\lambda,i)}$ is the optical thickness of nitrogen dioxide, $\tau_{(CO2\,\lambda,i)}$ is the optical thickness of carbon dioxide, $\tau_{(CH4\,\lambda,i)}$ is the optical thickness of methane, and $\tau_{(rel\,\lambda,i)}$ is the optical thickness due to Rayleigh scattering.

To obtain the optical thicknesses for gas absorption and scattering, we evaluated their dependencies as a function of the Julian day over the 2014–2020 period (Figure A1). Using these regression equations, we obtained the mean optical thickness of a particular gas for the i-day through linear interpolation. For testing, we compared the results obtained from our approach with the data of the third version of the AERONET for 2020 (Table A4). The comparisons demonstrated a difference within the accuracy of measurements of about 0.01 for the 380–875 nm interval. The largest difference (up to 0.03) was observed for $\tau_{aer\,\lambda}$ at 340 nm and for $\tau_{aer\,\lambda}$ at 1020 nm due to the deviation of the real water vapor content, which is important for $\tau_{aer\,1020}$, and due to the absence of the atmospheric pressure correction, which is important for $\tau_{aer\,340}$. However, we did not use $\tau_{aer\,\lambda}$ at these wavelengths in our analysis.

Since we need the aerosol optical thickness to attribute the aerosol effects on shortwave irradiance, we used only clear-sky conditions, which were chosen on the basis of hourly visual cloud observations in situations with 100% sunshine duration within the examined hour.

The main statistics for aerosol optical thickness at 500 nm ($\tau_{aer\ 500}$) and the Angstrom exponent within 440–870 nm over the 2021–2022 period are presented in Figure 2 together with their estimates over the 2001–2020 period. The Angstrom exponent is a parameter, which is determined as the slope of log dependencies between $\tau_{aer\ \lambda}$ and the wavelengths and is useful in attributing the particle size [33].

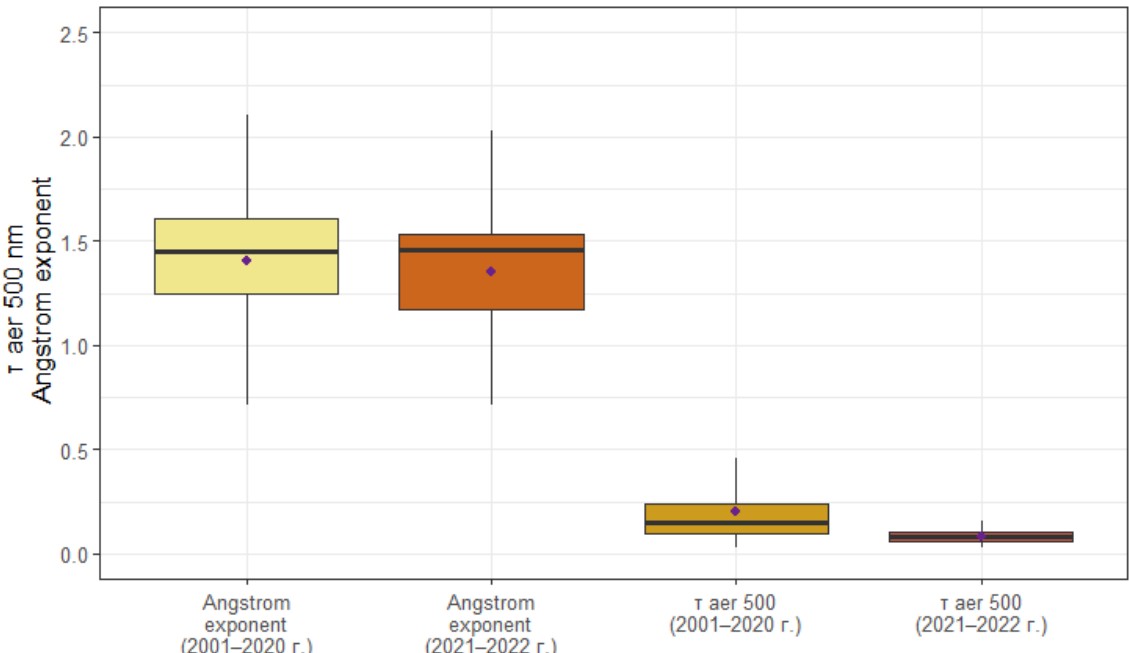

**Figure 2.** Box-and-whiskers diagram with the comparisons of Angstrom exponent 440–870 nm and aerosol optical thickness $\tau_{aer\ 500}$ over the 2021–2022 period and their climatological estimates over the 2001–2020 period (the MSU MO). The purple diamond indicates mean value, the line characterizes median, the box denotes interquartile range (from 1st to 3rd quartile), and whiskers show minimum and maximum values without outliers.

One can see that during the 2021–2022 period of the collocated measurements by the BSRN(MSU) complex and the CIMEL sun photometer, the atmosphere in Moscow was quite clean with the mean $\tau_{aer\ 500}$ of about 0.08, which is significantly smaller than its climatological estimate of about 0.2. So, this agrees with the whole tendency of brightening due to the decrease in aerosol content over the last decades in Moscow [35]. The Angstrom exponent 440–870 nm was close to the mean climatological value of about 1.4 over the whole period of AERONET observations since 2001.

Using the Angstrom parameter, we also evaluated aerosol optical thickness at 550 nm, which is also widely used [36]. The mean $\tau_{aer\ 550}$ during the 2021–2022 period is equal to 0.07, while the climatological value of $\tau_{aer\ 550}$ is about 0.17.

## 3. Results and Discussion

### 3.1. Factors Affecting Shortwave Irradiance

3.1.1. Aerosol Effects on Shortwave Irradiance in Snow and Snow-Free Clear-Sky Conditions

In order to evaluate the role of aerosol optical thickness on shortwave irradiance, the radiative data were divided into different $\tau_{aer\ 500}$ ranges. Figure 3 presents the dependence of direct, diffuse, global shortwave irradiances and net shortwave irradiance on the sine of solar elevation (sin h) separately for each range.

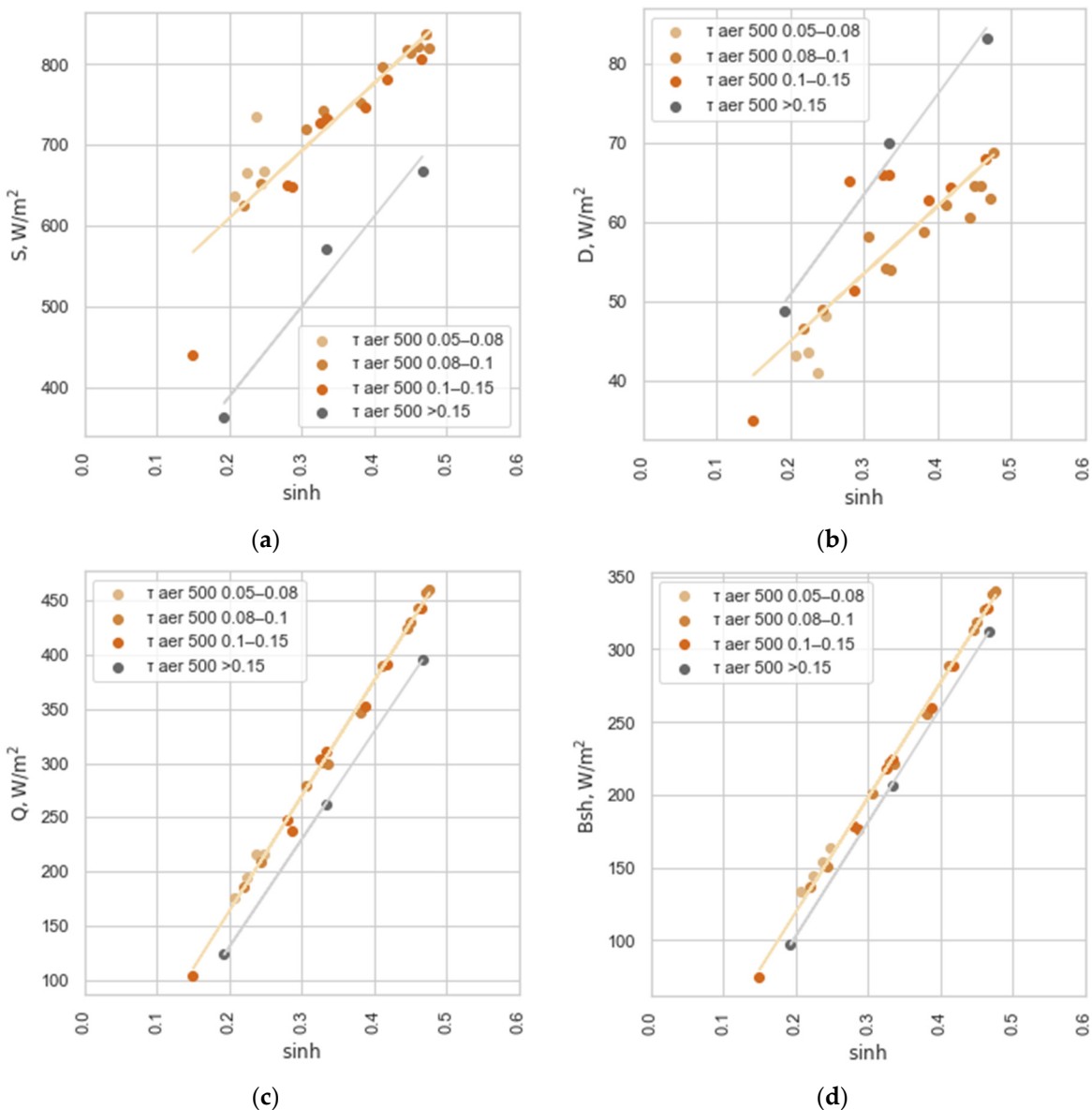

**Figure 3.** The dependence of direct (**a**), diffuse (**b**), global (**c**), and net shortwave irradiances (**d**) on sine of solar elevation (sin h) for different ranges of $\tau_{aer\ 500}$. Trend lines are shown for irradiances at $\tau_{aer\ 500} < 0.15$ and $\tau_{aer\ 500} > 0.15$. Snow-free clear-sky conditions. The MSU MO.

There is a noticeable attenuation of direct irradiance of about 150–200 W/m$^2$ with the increase of $\tau_{aer\ 500}$ (Figure 3a). On the contrary, the changes in diffuse irradiance are opposite due to the effects of aerosol scattering, however, they are not so large in the absolute magnitude (about 5–15 W/m$^2$). We also see the attenuation of global and net shortwave irradiances, but their decrease is smaller compared to direct irradiance due to the compensation by the increase in diffuse irradiance. These radiative changes due to $\tau_{aer\ 500}$ in clear-sky conditions are in agreement with the results obtained in other publications [24,37].

In order to quantify the loss of shortwave irradiance due to aerosol, we estimated the regression dependencies of global shortwave irradiance on $\tau_{aer\ 500}$ at different solar elevations (Table A5). Using these dependencies, we estimated the mean difference between Q at the observed $\tau_{aer\ 500}$ and Q in aerosol-free conditions ($\tau_{aer\ \lambda} = 0$) at different solar elevations.

Table 2 presents the calculated losses of global shortwave radiation due to $\tau_{aer\,\lambda}$. One can see that $\tau_{aer\,500}$ significantly affects global shortwave irradiance: from 8 to 36 W/m$^2$ at small solar elevation (h = 10°) up to 42–187 W/m$^2$ at h = 50°. Note, that our range of $\tau_{aer\,500}$ belongs to a relatively clean atmosphere, so we can speak about the effects of the background aerosol conditions.

**Table 2.** Mean losses of global shortwave irradiance (W/m$^2$) due to aerosol optical thickness $\tau_{aer\,500}$ at different solar elevations. Clear-sky snow-free conditions.

| Solar Elevation | 10° | | 20° | | 30° | | 40° | | 50° | |
|---|---|---|---|---|---|---|---|---|---|---|
| | W/m$^2$ | % | W/m$^2$ | % | W/m$^2$ | % | W/m$^2$ | % | W/m$^2$ | % |
| $\tau_{aer\,500}$ < 0.05 | 8.3 | 5.1 | 16.3 | 4.5 | 25.1 | 4.4 | 33.6 | 4.3 | 42 | 4.2 |
| $\tau_{aer\,500}$ 0.05–0.1 | 22.2 | 13.7 | 43.8 | 12.2 | 67.8 | 11.7 | 90.5 | 11.5 | 113.3 | 11.4 |
| $\tau_{aer\,500}$ 0.1–0.15 | 36.3 | 22.4 | 71.8 | 20.0 | 111.3 | 19.3 | 148.9 | 19.0 | 186.5 | 18.8 |

Since the levels of direct and diffuse irradiances have the opposite dependence on $\tau_{aer\,500}$, their ratio should be very sensitive to the changes in aerosol optical thickness. Figure 4 shows the direct to diffuse ratio as a function of $\tau_{aer\,500}$ in snow-free and snow conditions without cloudiness.

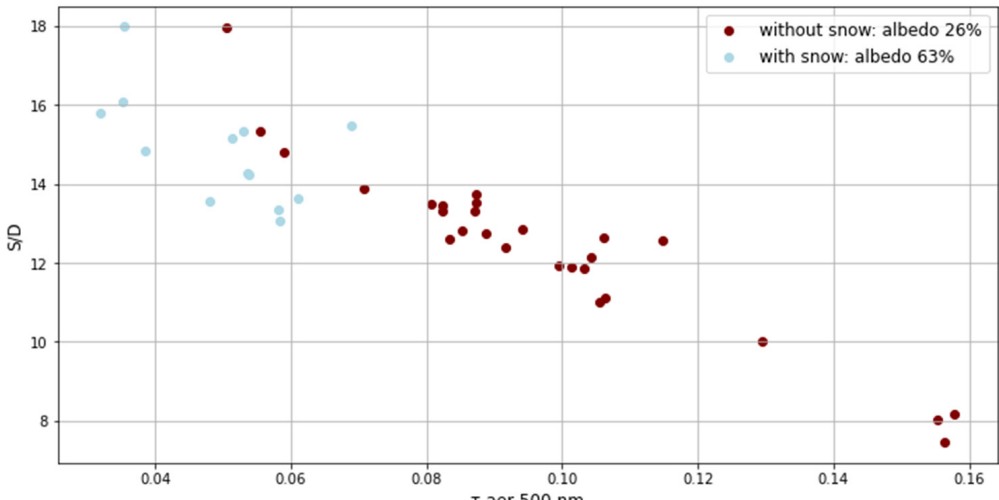

**Figure 4.** The ratio of direct to diffuse irradiances (S/D) as a function of $\tau_{aer\,500}$ in clear-sky conditions with and without snow. Note that in the sample without snow we had a large amount of clear-sky days in October. This provided higher mean surface albedo of about 26% due to the increased reflected irradiance from yellow grass and bare soil compared to grass conditions, when surface albedo is about 20%. The MSU MO.

The S/D ratio decreases from 15–18 to 8 with an increase in $\tau_{aer\,500}$ from 0.05 to 0.15 in snow-free conditions. However, in snow conditions, the ratio of direct to diffuse radiation is lower almost in all cases at similar $\tau_{aer\,500}$ due to the increase of diffuse irradiance because of additional multiple reflection from the surface. In both snow and snow-free conditions, we see the decrease in the S/D ratio as a function of $\tau_{aer\,\lambda}$.

The effects of surface albedo on global and net shortwave irradiance in clear-sky conditions are shown in Figure 5. One can see a noticeable increase in global shortwave irradiance during the winter, especially at high solar elevations (see Figure 5a). This happens due to a smaller aerosol loading during the cold period, which has a small impact on direct irradiance, while diffuse irradiance increases due to additional multiple reflection

from the surface. The difference may exceed 40 W/m² for relatively high solar elevations in winter conditions.

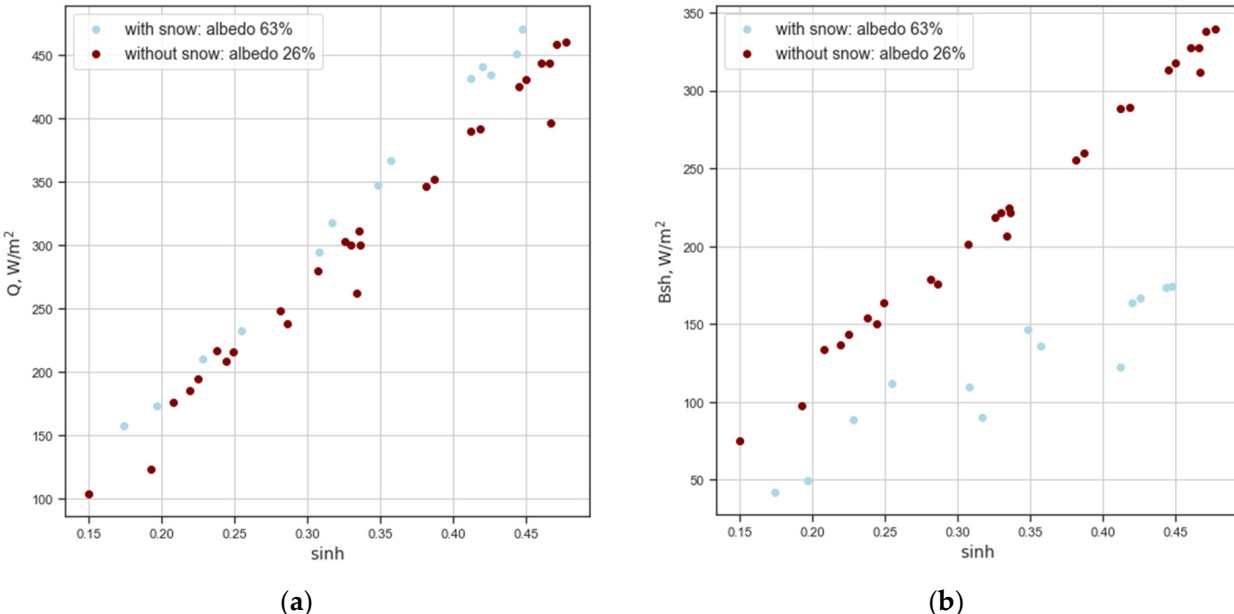

(**a**)                                                                                      (**b**)

**Figure 5.** Global shortwave irradiance Q (**a**) and net shortwave irradiance Bsh (**b**) as a function of sin h in clear-sky conditions with and without snow. MSU MO. We used the same cases as in Figure 4. The MSU MO.

However, the significant seasonal differences in water vapor content could be also important in attributing this bias. Using the CLIRAD(FC05)-SW radiative code [38], we estimated global shortwave irradiances with water vapor content of 0.3 cm and 2 cm, typical in winter and summer conditions [35]. The corresponding differences in global shortwave irradiance vary from 17.7 W/m² at solar elevation h = 10° to 50 W/m² at h = 30°. This means that low water vapor content in winter has also a noticeable contribution to the increase in global shortwave irradiance.

The difference in net shortwave irradiance (Figure 5b) is of the opposite sign due to a significant effect of high snow surface albedo on reflected irradiance. The difference reaches 150 W/m² in clear-sky conditions at maximum solar elevations observed in winter. The decrease in Bsh is only partly compensated by the increase of global shortwave irradiance due to smaller aerosol and water vapor contents and an additional increase in diffuse irradiance due to reflection.

### 3.1.2. Cloud Influence on Shortwave Irradiance

In order to evaluate the effects of cloudiness on shortwave irradiance, we used the cloud transmittance T(Q) = $Q/Q_0$, where Q is the global irradiance in cloudy conditions, and $Q_0$ in clear-sky conditions. The $Q_0$ values were obtained using the parameterizations, shown in Table A6 for snow and snow-free conditions. A similar procedure was applied for simulating the cloud transmittance for net shortwave irradiance T(Bsh). To characterize cloudiness, we used relative sunshine duration (Sd), where relative Sd = 1 means total absence of cloudiness and Sd = 0 means conditions with optically thick cloudiness. Note that 1 h averaging has been applied to the data, providing a better description of the cloud amount by the relative Sd parameter according to the ergodicity approach [39].

Figure 6 presents the dependence of T(Q) and T(Bsh) on the relative Sd for snow and snow-free conditions. For the determination of the type of underlying surface, we used standard meteorological data on snow cover and snow-cover height. We applied a filter of zero snow cover for snow-free conditions and a filter of 100% snow cover at snow height of more than 5 cm—to characterize snow conditions. The statistics of these characteristics are shown in Table 3.

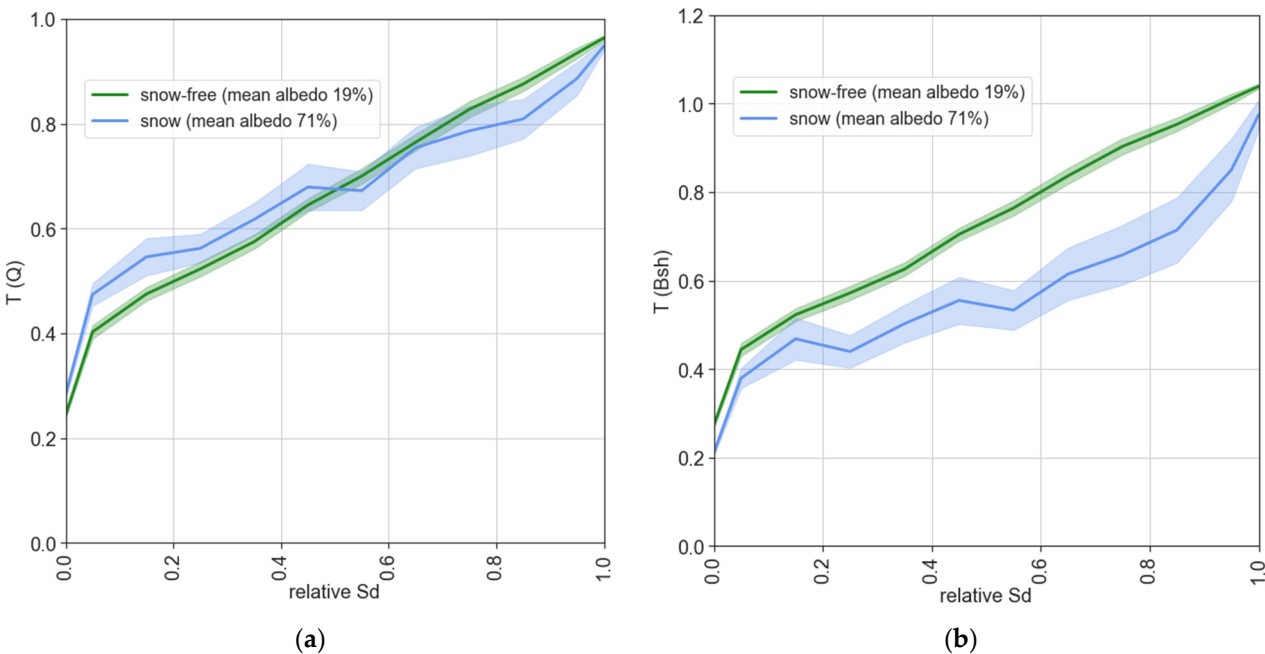

(**a**)                                                                    (**b**)

**Figure 6.** The dependence of cloud transmittances T(Q) (**a**) and T(Bsh) (**b**) on relative sunshine duration Sd in snow-free and snow conditions. Confidence intervals (95%) are shown by fill. The cases at low solar elevations (smaller than 5°) were removed from the sample. The MSU MO.

**Table 3.** The main statistics for cloud transmittances T(Q) and T(Bsh) in snow-free and snow conditions.

| The Relative Sd Intervals | Mean T(Q) | | Mean T(Bsh) | | Standard Deviation for T(Q) | | Standard Deviation for T(Bsh) | | Case Number | |
|---|---|---|---|---|---|---|---|---|---|---|
| | Snow-Free | Snow | Snow-Free | Snow | Snow-Free | Snow | Snow-Free | Snow | Snow-Free | Snow |
| 0 | 0.24 | 0.28 | 0.27 | 0.21 | 0.13 | 0.12 | 0.14 | 0.10 | 1516 | 1008 |
| 0–0.1 | 0.40 | 0.47 | 0.44 | 0.38 | 0.11 | 0.10 | 0.13 | 0.11 | 294 | 91 |
| 0.1–0.2 | 0.48 | 0.55 | 0.52 | 0.47 | 0.10 | 0.11 | 0.11 | 0.15 | 213 | 40 |
| 0.2–0.3 | 0.52 | 0.56 | 0.57 | 0.44 | 0.10 | 0.08 | 0.11 | 0.10 | 183 | 30 |
| 0.3–0.4 | 0.57 | 0.62 | 0.63 | 0.50 | 0.09 | 0.08 | 0.11 | 0.11 | 182 | 24 |
| 0.4–0.5 | 0.64 | 0.68 | 0.70 | 0.56 | 0.09 | 0.11 | 0.11 | 0.14 | 212 | 26 |
| 0.5–0.6 | 0.70 | 0.67 | 0.76 | 0.53 | 0.11 | 0.09 | 0.13 | 0.11 | 227 | 25 |
| 0.6–0.7 | 0.76 | 0.75 | 0.84 | 0.61 | 0.12 | 0.11 | 0.14 | 0.16 | 229 | 28 |
| 0.7–0.8 | 0.83 | 0.79 | 0.90 | 0.66 | 0.12 | 0.11 | 0.13 | 0.16 | 209 | 22 |
| 0.8–0.9 | 0.88 | 0.81 | 0.95 | 0.71 | 0.11 | 0.10 | 0.13 | 0.19 | 274 | 26 |
| 0.9–1 | 0.93 | 0.89 | 1.01 | 0.85 | 0.11 | 0.13 | 0.12 | 0.29 | 430 | 64 |
| 1 | 0.96 | 0.95 | 1.04 | 0.98 | 0.09 | 0.07 | 0.10 | 0.27 | 1229 | 238 |

Figure 6 shows a strong nonlinear dependence of T(Q) on relative sunshine duration, especially when the T(Q) values are close to zero. We see a significant increase of T(Q) from 0.24 to 0.40 in summer and from 0.28 to 0.47 in winter, when relative Sd values deviate only slightly from zero. At high relative Sd close to 1.0, the instant T(Q) and T(Bsh) values

may exceed 1 (see high standard deviations). This happens in conditions with broken cloudiness due to 3D scattering from lateral cloud sides, which provides an additional increase in diffuse irradiance, while in cloud gaps no attenuation of direct irradiance is observed. When the relative Sd is equal to 1, mean cloud transmittance T(Q) is slightly smaller than 1, since thin cirrus clouds do not affect sunshine duration, but attenuate the global shortwave irradiance.

The influence of cloudiness on solar irradiance depends on cloud types [40]. The application of relative Sd as a proxy of cloud characteristic works only for semi-transparent clouds, but when cloudiness is optically thick, relative Sd is equal to zero for different cloud types. This provides an increase in standard deviations, shown in Table 3. However, since cloud transmittance has a pronounced dependence on relative Sd, and the latter is a very simple and automatically measured parameter, the obtained dependencies could be useful in the analysis of cloud effects.

Quite interesting nonlinear dependence was obtained for T(Bsh) in winter conditions. One can see that the T(Bsh) values are significantly lower than the T(Q) at relative Sd = 0.25–0.9. This may occur due to a significant increase in reflected irradiance in broken cloud conditions due to multiple scattering from the surface. Larger T(Bsh) deviations in winter compared to those in summer, can be explained by the significant influence of surface albedo and its variability.

The analysis of radiative effects of the atmospheric factors, described in this section, allows us to better understand the reasons for shortwave irradiance variability shown in Section 3.2.

### 3.2. Radiative Regime at the MSU MO According to the RAD-MSU(BSRN) Measurements

Figure 7 presents monthly doses of net longwave and net shortwave irradiances, sunshine duration, monthly mean D/Q ratios, and surface albedo over the whole period of measurements. Since Moscow is located at 55.7° N, the changes in solar elevation h and in duration of the daylight period with h > 0° are the key reasons, responsible for the seasonal variability of the net shortwave irradiance. The net shortwave irradiance varies dramatically during the seasons, but is always positive, while the net longwave irradiance is always negative. Both components of the net irradiance are close to zero in winter. In summer, the net shortwave irradiance dominates in the absolute magnitude over the net longwave irradiance, while in winter we see the opposite picture.

In winter, the net shortwave irradiance is close to zero due to smaller solar elevation, shorter daylight duration, higher occurrence of cloudy conditions [41] and high surface albedo. The prevailing cloudy conditions in winter can be seen from the high D/Q ratio which is close to 1 (see Figure 7). A similar tendency is typical in moderate climates of Eastern Europe with cyclone weather, prevailing in winter [42]. In addition, snow cover provides strong reflection and the Bsh decreases. This decrease is only partly compensated by a smaller aerosol optical thickness and water vapor content in winter conditions (see the Section 3.1.1). The net longwave irradiance in winter, on the contrary, is higher due to the low surface temperatures compared to those in summer conditions. However, their monthly doses are still below zero.

Figure 8 presents the dependence of the daily doses of net shortwave irradiance and the net longwave irradiance on the daily sums of sunshine duration in snow-free and snow conditions. We determined these conditions similar to the procedure used in the analysis of cloud transmittance. The daily Sd sums describe both the changes in daylight period with h > 0, and cloudiness. Hence, for net shortwave irradiance these factors provide a positive dependence on the Sd sums. For net longwave irradiance, on the contrary, the larger sums of sunshine duration lead to higher negative values due to the large contrast in temperature between the surface and the atmosphere, while the daylight period is not important.

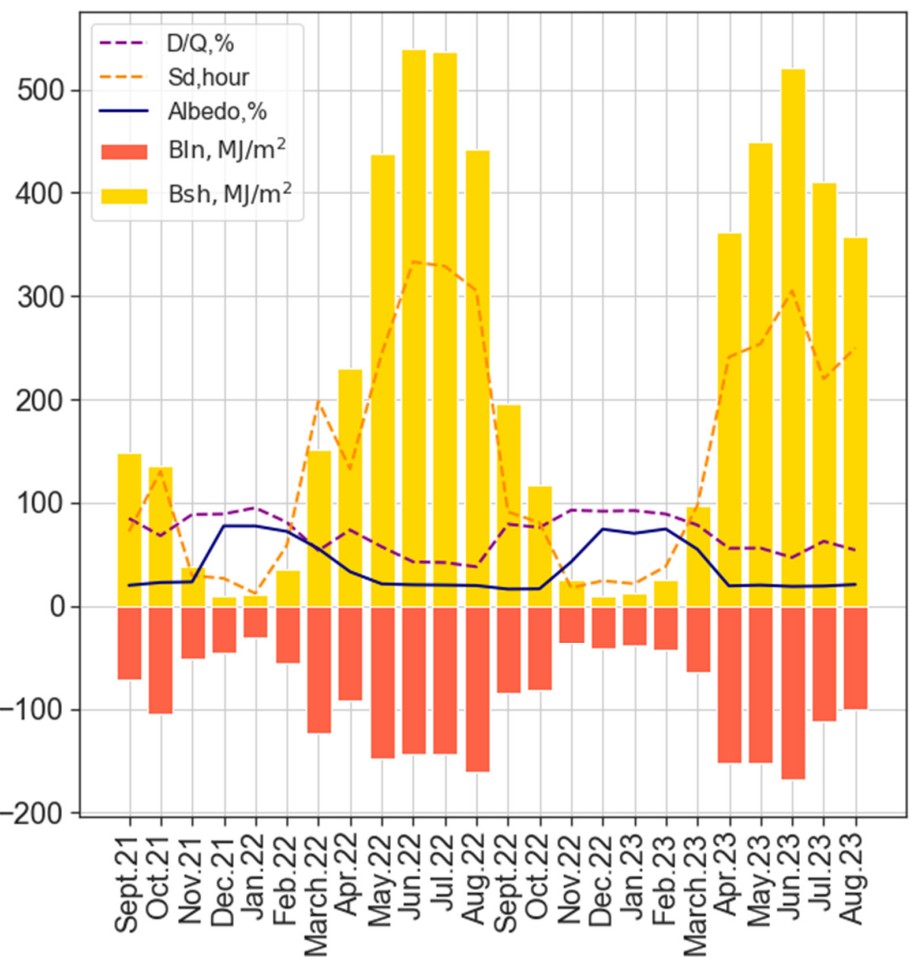

**Figure 7.** Monthly sums of sunshine duration Sd (hours), net shortwave (Bsh) and net longwave (Bln) irradiances (MJ/m$^2$), monthly mean ratio of diffuse to global shortwave irradiances (D/Q), and surface albedo (%) over the whole period of measurements. The MSU MO.

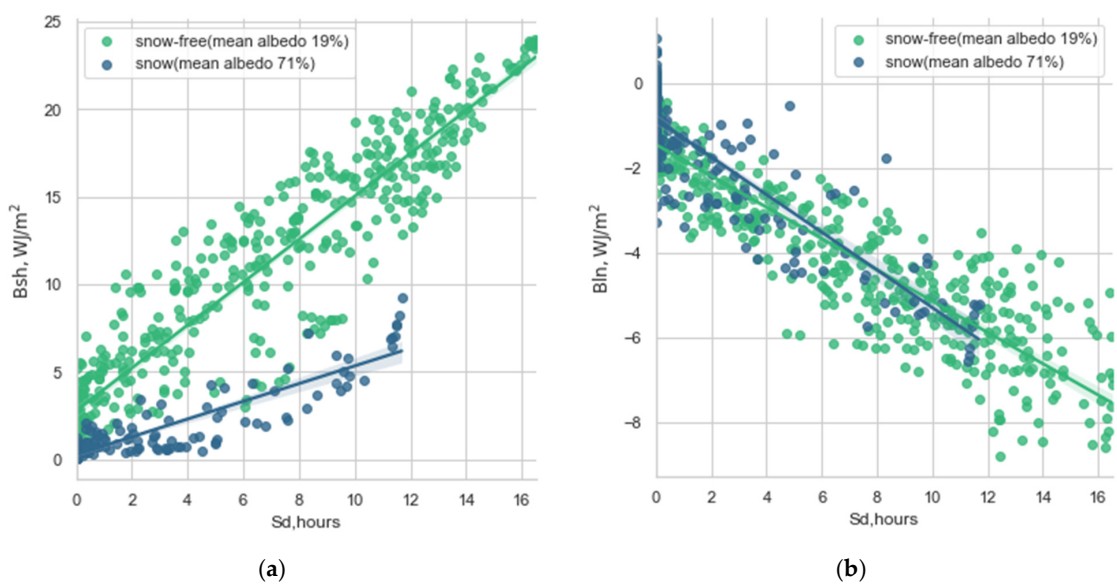

**Figure 8.** Daily doses of net shortwave (**a**) and net longwave (**b**) irradiances as a function of daily sums of sunshine duration in snow-free and snow conditions. The MSU MO.

### 3.3. The Comparisons of the Radiative Regime during the 2021–2023 Period with Long-Term Observations

Figure 9 shows the mean seasonal doses of global shortwave irradiance, sunshine duration, net shortwave irradiance from the RAD-MSU(BSRN) measurements, and mean total cloud amount N for the September 2021–August 2023 period of observations. They are compared with the climatological values over the 1955–2020 period from the standard observations [32].

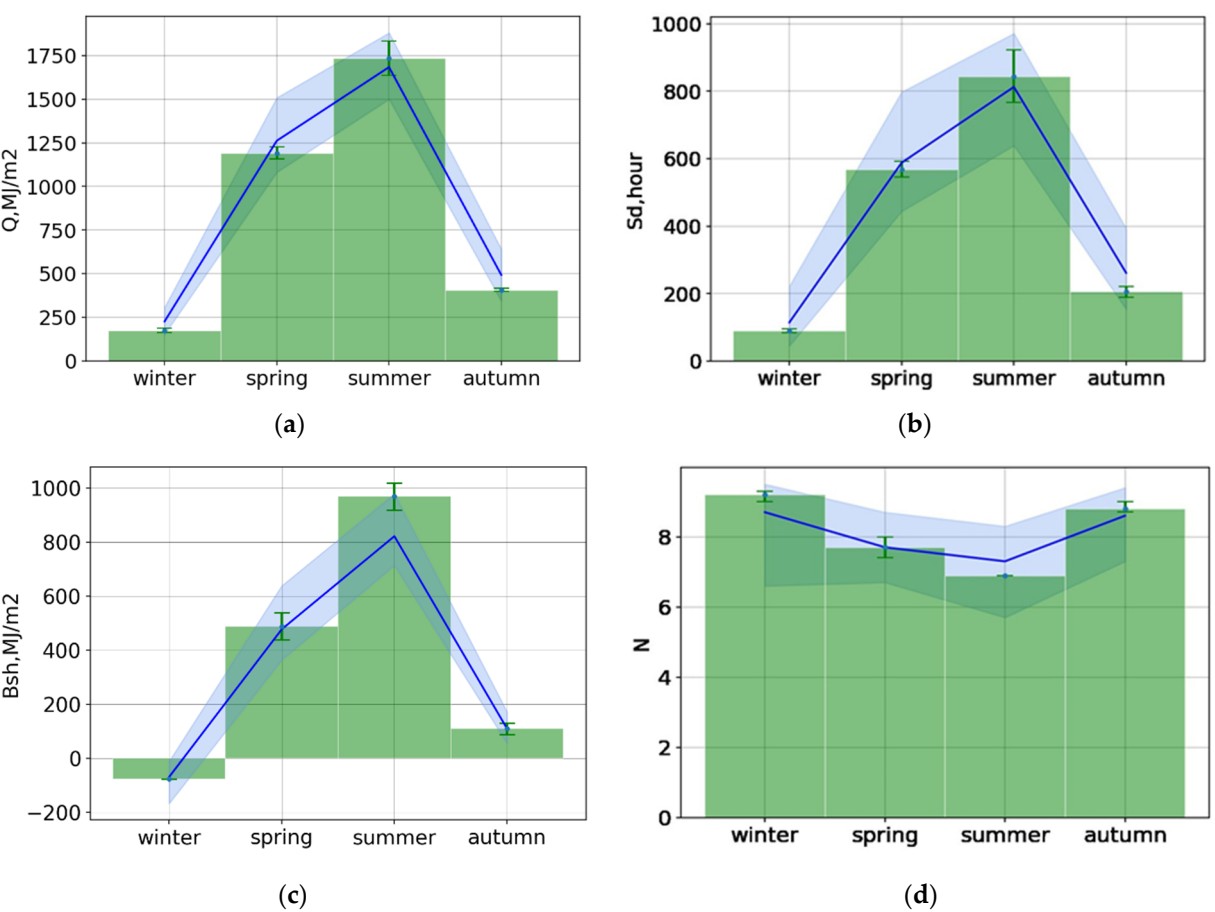

**Figure 9.** Mean seasonal doses of global shortwave irradiance (**a**), sunshine duration (**b**), net short-wave irradiance (**c**) according to the RAD-MSU(BSRN) measurements and total cloud amount visual observations (in tenth) N (**d**) over the September 2021–August 2023 period (green histograms with error bars), and the results of long-term observations over the 1955–2020 period (blue lines with error bars). Error bars depict min/max doses in both cases. Cloud amount N is obtained from 1 h cloud visual observations over the whole period of measurements since 1954. The MSU MO.

Seasonal doses of global shortwave irradiance for the 2021–2023 period were characterized by lower values during the cold period. The anomalies relative to the long-term average doses were −22% and −17%, respectively, for the winter and autumn periods. The spring period was also characterized by a negative anomaly of about −5.5%. In summer, on the contrary, due to smaller total cloud amount N, the positive anomalies of about +3% for global shortwave irradiance and +18% for net shortwave irradiance were observed. The higher increase of Bsh compared to Q against climatological values may be explained, to some extent, by the lower reflected irradiance obtained by the RAD-MSU(BSRN) compared to the standard measurements, which were discussed above.

We should note that the increase of global shortwave irradiance in summer over the last years is a typical trend found over the whole European territory [2,28]. The negative anomalies of global shortwave irradiance in winter have also been forecasted by the chemical climate models at high latitudes in the 21st century [43]. So, we could state that the tendencies reported by the RAD-MSU(BSRN) measurements, reflect the main features of changing climate.

## 4. Conclusions

The RAD-MSU(BSRN) complex provides the high-quality measurements, which allow us to characterize the Moscow radiative climate in more detail, compared to previous assessments. At the same time, the comparisons with shortwave radiation from the standard radiative observations, provided by the Russian radiative instruments, demonstrate a satisfactory agreement. The differences mainly lie within the limits of instrumental uncertainties.

In clear-sky snow-free conditions, even at low aerosol loading ($\tau_{aer,500} < 0.2$), the loss of global shortwave irradiance varied as a function of solar elevation from 37 to 186 W/m$^2$ comprising about 18–22%, with a significant decrease in direct irradiance and an increase in diffuse irradiance due to multiple scattering.

In the absence of cloudiness, the S/D ratio decreased from 15–18 to 8 with an increase in $\tau_{aer\ 500}$ from 0.05 to 0.15 in snow-free conditions. However, in snow conditions, the ratio of direct to diffuse radiation was lower almost in all cases at similar $\tau_{aer\ 500}$, due to the increase of diffuse irradiance because of multiple reflection from surface.

In winter, in clear-sky conditions high snow surface albedo together with smaller aerosol and water vapor content provided an increase in global shortwave irradiance of about 45 W/m$^2$ (or 9%) at h = 30°. At the same time, net shortwave irradiance demonstrated a significant decrease due to prevailing effects of reflected irradiance at high snow albedo.

The nonlinear dependences of the Q and Bsh cloud transmittance on relative sunshine duration were obtained. We showed a significant T(Q) increase from 0.24 to 0.4 in summer and from 0.28 to 0.47 in winter, when relative Sd values only slightly deviated from zero. Similar changes were observed for the Bsh cloud transmittance.

Mean seasonal changes in global shortwave irradiance during the 2021–2023 period compared to the climatological mean values over the 1955–2020 period, were characterized by negative anomalies (−22%) in winter due to the cloud amount increase, which was reflected in the reduced relative Sd. In summer, positive anomalies in global shortwave irradiance were observed (+3%) due to cloud amount reduction. This is in line with the global tendencies in the long-term changes of global shortwave irradiance in moderate climates of Europe over the last years.

**Author Contributions:** N.C.: Conceptualization, Methodology, Supervision, Editing; D.P.: Data curation, Visualization, Validation, Investigation; Writing—Original draft preparation, Editing; A.P.: Software, Visualization, Investigation, Validation, Methodology, Editing; E.Z.: Visualization, Investigation, Methodology, Editing. All authors have read and agreed to the published version of the manuscript.

**Funding:** The study of comparisons between the RAD-MSU(BSRN) measurements and standard radiative observations and simulation of radiative effects of albedo and water vapor were supported by the Russian Science Foundation project No. 23-77-01030. The work was carried out within the framework of the work of the MSU Collective Use Center (Monitoring of Atmospheric Radiation, No. 460191494). The comparisons with long-term measurements and studies of aerosol effects were supported by grant No. 075-15-2021-574. The evaluation of cloud radiative effects was supported by the Ministry of Education and Science of the Russian Federation as part of the program of the Moscow Center for Fundamental and Applied Mathematics under agreement No. 075-15-2022-284.

**Institutional Review Board Statement:** Not applicable.

**Informed Consent Statement:** Not applicable.

**Data Availability Statement:** The results presented in the manuscript were created according to the RAD-MSU(BSRN) observations (available on request, the data are not publicly available due to internal policy of MSU and MSU Collective Use Center), the AERONET data (https://aeronet.gsfc.nasa.gov/, last access 30 July 2023), and the CIMEL sun photometer data (available on request). Figures were created using the Python version 3.8.5 (available at https://www.python.org, last access on 18 October 2022) and the R version 4.3.1 (available at https://cran.r-project.org/, last access 15 October 2023).

**Conflicts of Interest:** The authors declare no conflict of interest.

## Appendix A

**Table A1.** Monthly mean absolute (MJ/m$^2$) and relative (%) differences in direct, diffuse, reflected, and global shortwave irradiances between the measurements of the MSU-RAD(BSRN) complex and the standard measurements. The MSU MO. 2022.

|  | Direct Irradiance | Diffuse Irradiance | Reflected Irradiance | Global Irradiance |
|---|---|---|---|---|
|  | $\Delta/\Delta\%$ | $\Delta/\Delta\%$ | $\Delta/\Delta\%$ | $\Delta/\Delta\%$ |
| January | 0/0 | −5.2/−12.7 | 3.3/9.7 | −5.2/−11.7 |
| February | 6.5/5.3 | −3.6/−4.7 | 0.1/0.1 | −1.8/−1.5 |
| March | 25.5/5.1 | −3.6/−2.9 | 5.6/3.3 | 5.7/1.8 |
| April | 3.5/1.4 | −8.9/−4.4 | −0.6/−0.6 | −6.9/−2.1 |
| May | −0.5/−0.1 | −4.4/−1.6 | −2.9/−2.3 | −3.9/−0.7 |
| June | 0.8/0.1 | −2.4/−1 | −5.1/−3.4 | −11.4/−1.7 |
| July | 16.4/2.4 | 0.5/0.2 | −8.9/−6.4 | 8.7/1.3 |
| August | −4.2/−0.7 | −2.9/−1.2 | −10.3/−9.3 | −4.3/−0.8 |
| October | 1.6/1 | −4.6/−5.3 | −2.3/−9.2 | −5.4/−3.9 |
| November | 0.9/2.7 | −2.5/−7.1 | −0.8/−4.9 | −3/−7.1 |
| December | −0.1/−0.2 | −2.7/−10.3 | −1.1/−4.8 | −3.1/−9.5 |
| Year | 50.4/1.4 | −40.3/−2.5 | −23/−2.4 | −30.5/−0.9 |

Note: the data in September are absent due to the absence of standard measurements.

**Table A2.** Monthly doses (MJ/m$^2$) of direct, diffuse, reflected, and global shortwave irradiances, downwelling and upwelling longwave irradiances, and net irradiance according to the new MSU-RAD(BSRN) complex at the MSU MO.

|  | Direct Irradiance | Diffuse Irradiance | Reflected Irradiance | Global Irradiance | Downwelling Longwave Irradiance | Upwelling Longwave Irradiance | Net Irradiance |
|---|---|---|---|---|---|---|---|
| September, 21 | 47.2 | 139.3 | 38.1 | 186.5 | 869.8 | 941.2 | 77.1 |
| October, 21 | 90.2 | 89.1 | 39.9 | 179.3 | 807.5 | 912.8 | 30.5 |
| November, 21 | 13.4 | 36.4 | 12.0 | 49.8 | 783.2 | 835.2 | −14.3 |
| December, 21 | 6.9 | 29.5 | 26.9 | 36.4 | 696.4 | 742.9 | −37.0 |
| January, 22 | 3.4 | 41.1 | 33.7 | 44.5 | 732.0 | 763.1 | −20.3 |
| February, 22 | 32.4 | 78.5 | 75.3 | 110.9 | 668.3 | 724.2 | −20.2 |
| March, 22 | 192.9 | 126.5 | 167.2 | 319.5 | 670.5 | 793.7 | 29.4 |
| April, 22 | 133.8 | 200.0 | 103.9 | 333.8 | 790.8 | 883.5 | 137.2 |
| May, 22 | 278.7 | 283.7 | 122.7 | 562.4 | 836.5 | 983.9 | 292.3 |
| June, 22 | 451.7 | 247.1 | 142.8 | 698.8 | 909.4 | 1049.4 | 394.2 |
| July, 22 | 411.4 | 264.8 | 139.8 | 676.2 | 972.5 | 1116.3 | 392.6 |
| August, 22 | 430.3 | 245.9 | 110.1 | 676.2 | 983.6 | 1143.9 | 282.4 |

**Table A2.** *Cont.*

| | Direct Irradiance | Diffuse Irradiance | Reflected Irradiance | Global Irradiance | Downwelling Longwave Irradiance | Upwelling Longwave Irradiance | Net Irradiance |
|---|---|---|---|---|---|---|---|
| September, 22 | 76.6 | 155.5 | 40.4 | 232.1 | 863.3 | 947.9 | 107.1 |
| October, 22 | 54.3 | 87.2 | 24.9 | 141.5 | 847.2 | 929.8 | 34.0 |
| November, 22 | 7.7 | 35.0 | 16.9 | 42.7 | 768.9 | 805.5 | −10.9 |
| December, 22 | 6.4 | 26.3 | 22.8 | 32.7 | 737.3 | 778.7 | −31.4 |
| January, 23 | 5.8 | 31.4 | 25.1 | 37.2 | 734.5 | 773.0 | −26.3 |
| February, 23 | 19.9 | 71.7 | 66.5 | 91.6 | 655.7 | 698.9 | −18.1 |
| March, 23 | 76.9 | 133.4 | 111.5 | 210.2 | 765.1 | 830.5 | 33.5 |
| April, 23 | 234.4 | 218.3 | 90.6 | 452.8 | 784.6 | 937.4 | 209.4 |
| May, 23 | 302.1 | 266.6 | 119.5 | 568.7 | 852.8 | 1004.6 | 297.4 |
| June, 23 | 394.4 | 255.1 | 129.1 | 649.5 | 872.0 | 1040.2 | 352.3 |
| July, 23 | 238.3 | 276.3 | 103.2 | 514.6 | 979.9 | 1091.8 | 299.5 |
| August, 23 | 261.4 | 212.7 | 104.4 | 474.1 | 987.7 | 1091.5 | 265.9 |

**Table A3.** Calibration constants $S_{o.\lambda}$ at different wavelengths (courtesy of Dr. T. Eck. NASA GSFC).

| Wavelengths | 1020 nm | 870 nm | 670 nm | 440 nm | 500 nm | 380 nm | 340 nm |
|---|---|---|---|---|---|---|---|
| Calibration constants, $S_{o.\lambda}$ | 13,902 | 19,880 | 24,785 | 18,666 | 15,967 | 36,550 | 39,394 |

**Table A4.** Differences in total $\tau$ ($\Delta\tau$) and in $\tau_{aer}$ ($\Delta\tau_{aer}$) estimated in this work and in the AERONET algorithm at different wavelengths. The MSU MO. 2020.

| Wavelengths | 340 nm | 380 nm | 440 nm | 500 nm | 675 nm | 870 nm | 1020 nm |
|---|---|---|---|---|---|---|---|
| | | | | $\Delta\tau$ | | | |
| Mean | 0.0023 | 0.0015 | 0.0014 | 0.0018 | 0.0000 | −0.0002 | −0.0090 |
| Max | 0.0258 | 0.0184 | 0.0136 | 0.0116 | 0.0056 | 0.0032 | 0.0132 |
| Min | −0.0092 | −0.0067 | −0.0045 | −0.0035 | −0.0028 | −0.0024 | −0.0372 |
| Standard deviation | 0.0049 | 0.0035 | 0.0026 | 0.0025 | 0.0014 | 0.0013 | 0.0070 |
| | | | | $\Delta\tau_{aer}$ | | | |
| Mean | −0.0009 | −0.0008 | 0.0026 | 0.0022 | 0.0001 | −0.0001 | −0.0088 |
| Max | 0.0331 | 0.0228 | 0.0174 | 0.0131 | 0.0066 | 0.0034 | 0.0113 |
| Min | −0.0172 | −0.0108 | −0.0040 | −0.0028 | −0.0025 | −0.0025 | −0.0357 |
| Standard deviation | 0.0071 | 0.0047 | 0.0032 | 0.0028 | 0.0016 | 0.0013 | 0.0068 |

**Table A5.** Parametrizations of global solar irradiance (Q W/m$^2$) and net shortwave irradiance (Bsh W/m$^2$) on sine of solar elevation (sin h) and aerosol optical thickness ($\tau_{aer,500}$) for snow-free conditions. The MSU MO.

| a. Q Dependence on sin h for Different $\tau_{aer\ 500}$ | | |
|---|---|---|
| $\tau_{aer}$ range | Q | $R^2$ |
| <0.05 | 19.148 $\times$ sin h − 43.074 | 1 |
| 0.05–0.08 | 19.064 $\times$ sin h − 49.633 | 0.99 |
| 0.08–0.1 | 17.596 $\times$ sin h − 41.954 | 1 |
| 0.1–0.12 | 17.804 $\times$ sin h − 50.009 | 1 |
| >0.12 | 16.208 $\times$ sin h − 41.939 | 0.98 |

**Table A5.** *Cont.*

| b. Q dependence on $\tau_{aer}$ for different h | | |
|---|---|---|
| **h. °** | **Q** | **$R^2$** |
| 10 | $161.75 \times e^{-2.118 \times \tau aer}$ | 1 |
| 20 | $359.63 \times e^{-1.858 \times \tau aer}$ | 0.97 |
| 30 | $577.53 \times e^{-1.787 \times \tau aer}$ | 0.95 |
| 40 | $785.43 \times e^{-1.754 \times \tau aer}$ | 0.93 |
| 50 | $993.33 \times e^{-1.735 \times \tau aer}$ | 0.93 |

**Table A6.** Q and Bsh dependences on sin h for snow and snow-free conditions. The MSU MO.

| | Q | | Bsh | |
|---|---|---|---|---|
| | **Q** | **$R^2$** | **Bsh** | **$R^2$** |
| Summer (grass, snow-free surface) | $1063.2 \times \sin h - 49.498$ | 1 | $788.35 \times \sin h - 38.755$ | 1 |
| Winter (snow surface) | $1161.2 \times \sin h - 53.915$ | 1 | $434.33 \times \sin h - 23.579$ | 0.87 |

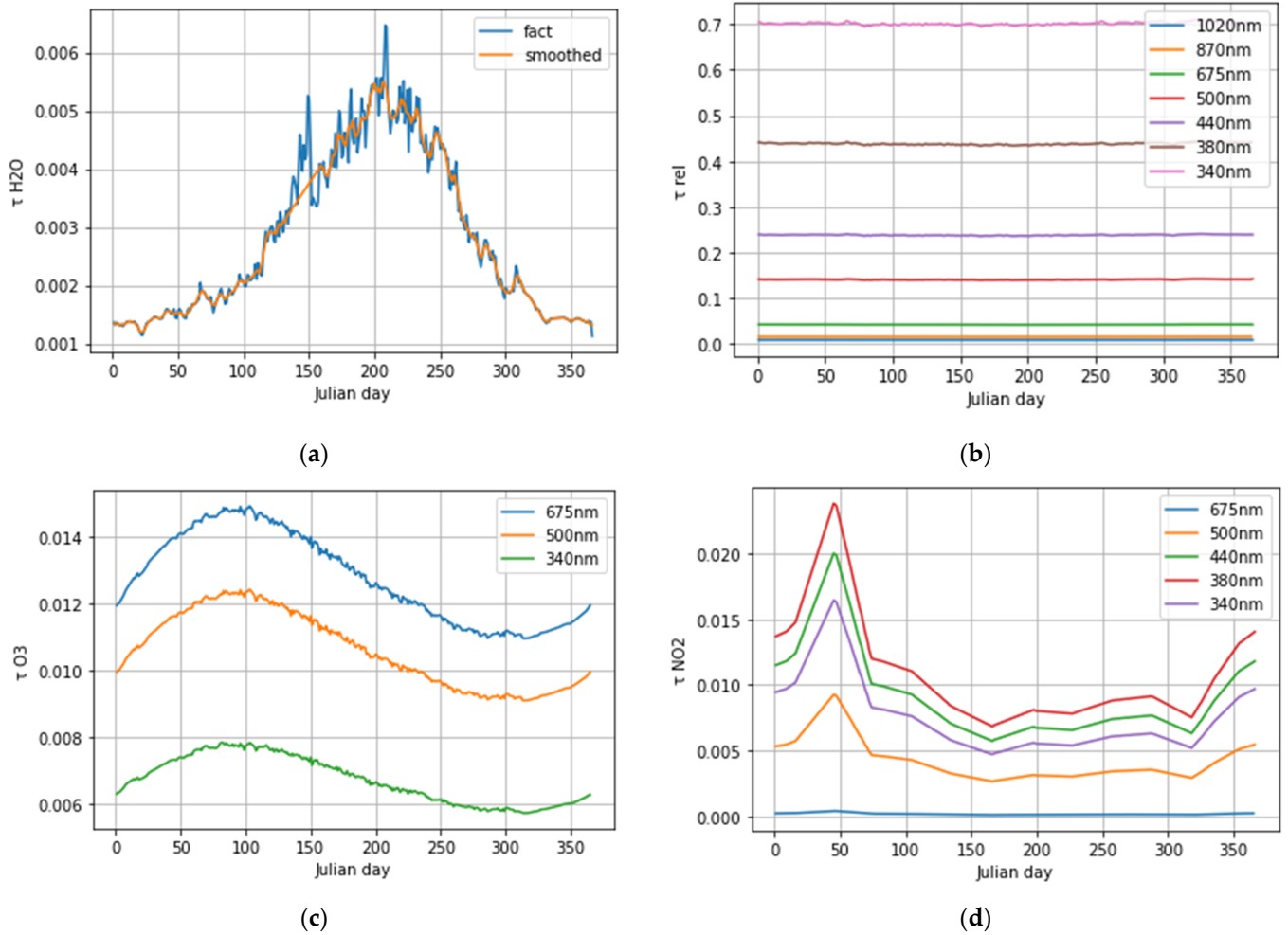

**Figure A1.** Annual changes in optical thickness of water vapor (**a**), Rayleigh scattering (**b**), ozone (**c**) and nitrogen dioxide (**d**) at different wavelengths. 2014–2020. Moscow.

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
