# Peer review of "Radiative Regime According to the New RAD-MSU(BSRN) Complex in Moscow: The Roles of Aerosol, Surface Albedo, and Sunshine Duration"

_atmosphere, doi:10.3390/atmos15020144_

Round 1

Reviewer 1 Report

Comments and Suggestions for Authors

The article provides new findings about the radiation regime in Moscow based on the new instrumentation complex MSU-RAD(BSRN).

I can acknowledge that the authors have put efforts on the scientific work, but the presentation needs a clear improvement before an acceptance can be considered. From my point of view, the manuscript is not easy to read because there is not a clear path on what the goals are, and how these goals are reached.

Formatting needs improvement. Mathematical formulations (e.g. for tau) are sometimes displayed incorrectly. Table 1 goes outside the horizontal bounds of the page. Equation 1 is also misaligned.

Some of the manuscript figures should be regenerated. Figure 4 is significantly pixelised. Figure 5 has the same problem, but it is more subtle because the graphs are shown smaller.

Comments on the Quality of English Language

Moderate English editing is required. Lack of commas and missing articles are quite spread throughout the text. Use of wrong terms can also be seen (e.g. Line 183, "qualify" instead of "quantify").

Author Response

Thank you very much for taking the time to review this manuscript. Please see the attachment.

Reviewer 2 Report

Comments and Suggestions for Authors

This research comprehensively analyzes surface radiation for Moscow from a BSRN station. The paper looks good to me, but I suggest a few minor revisions or clarifications.

1.       I suggest adding a map to show the site location. Some critical information was not listed, such as the site Lat, Lon, Elevation, Land cover type, etc.

2.       Many research focus on AOD 550nm, like the MODIS and VIIRS aerosol products. Is there any particular reason that the authors use AOD 500 as the major statistics?

3.       Fig 4~5 shows non snow albedo of 0.26? It looks pretty large for the snow-free surface. Any more clues, such as the land cover type?

4.       Fig 7-8 albedo 0.4? similar to the previous one, any more details?

5.       Fig 9, what is N of cloud amount? I did not quite get that, could you elaborate it?

Author Response

(The authors gave the same response as above.)

Reviewer 3 Report

Comments and Suggestions for Authors

Review of “Radiative regime according to the new MSU-RAD(BSRN) complex in Moscow: the role of aerosol, surface albedo and sunshine duration” by Piskunova et al.

This study utilized two-year observations to examine the radiative climate in Moscow and aimed to identify the potential factors impacting radiation. The recent two-year observations are validated by the long-term measurements. The materials presented in this paper are interesting and well-suited to the scope of the current journal. It is recommended to be accepted after the authors revise the manuscript by reflecting on the following comments.

Major concerns:

The structure of Sect. 3 would be better if the authors first compare the MSU-RAD (BSRN) measurements with long-term observations (Sect. 3.3), then show the annual cycle of radiation (Sect. 3.2), and finally discuss the factors (Sect. 3.1).

Specific comments:

Line 82, the location (latitude and longitude) of the station should be specified.

Line 177, more details should be discussed, e.g., how different aerosol loadings affect global/direct/diffuse radiation? Some references are recommended to be cited here, e.g., doi: 10.1007/s00376-019-9010-4.

Line 178, how do the authors derive this value?

Line 223, cloud types are significant in impacting radiation, e.g., doi: 10.5194/acp-23-8169-2023, which should be discussed here.

Line 275, it should be moved to Line 272.

Line 278, small -> low

Line 286, I do not actually understand what ‘the length of the day’ means, is it the same as sunshine duration? If it is the sunshine duration, this sentence might be wrong.

Line 288, the separation of albedo into two categories requires further elaboration regarding its aim; otherwise, it may lead to confusion.

Line 309, the relatively less cloud amount is also the factor for the larger net SW radiation.

Line 310, it seems, in summer, the net shortwave irradiance increases more than global shortwave irradiance relative to the past periods, which could be discussed here.

Author Response

(The authors gave the same response as above.)

Round 2

Reviewer 1 Report

Comments and Suggestions for Authors

Please find my comments in the attachment.

Comments on the Quality of English Language

Please find my comments about English in the previous attachment.

Author Response

Thank you for the review, you can see our response in the file below

Reviewer 2 Report

Comments and Suggestions for Authors

The revised manuscript looks good to me.

Author Response

Thank you very much!